# A Combined Image- and Coordinate-Based Meta-Analysis of Whole-Brain Voxel-Based Morphometry Studies Investigating Subjective Tinnitus

**DOI:** 10.3390/brainsci12091192

**Published:** 2022-09-03

**Authors:** Punitkumar Makani, Marc Thioux, Sonja J. Pyott, Pim van Dijk

**Affiliations:** 1Department of Otorhinolaryngology–Head and Neck Surgery, University of Groningen, University Medical Center Groningen, P.O. Box 30.001, 9700 RB Groningen, The Netherlands; 2Graduate School of Medical Sciences (Research School of Behavioral and Cognitive Neurosciences), University of Groningen, FA30, P.O. Box 196, 9700 AD Groningen, The Netherlands

**Keywords:** combined image- and coordinate-based meta-analysis, hearing loss, magnetic resonance imaging, tinnitus, voxel-based morphometry

## Abstract

Previous voxel-based morphometry (VBM) studies investigating tinnitus have reported structural differences in a variety of spatially distinct gray matter regions. However, the results have been highly inconsistent and sometimes contradictory. In the current study, we conducted a combined image- and coordinate-based meta-analysis of VBM studies investigating tinnitus to identify robust gray matter differences associated with tinnitus, as well as examine the possible effects of hearing loss on the outcome of the meta-analysis. The PubMed and Web of Science databases were searched for studies published up to August 2021. Additional manual searches were conducted for studies published up to December 2021. A whole-brain meta-analysis was performed using Seed-Based d Mapping with Permutation of Subject Images (SDM-PSI). Fifteen studies comprising 423 individuals with tinnitus and either normal hearing or hearing loss (mean age 50.94 years; 173 females) and 508 individuals without tinnitus and either normal hearing or hearing loss (mean age 51.59 years; 234 females) met the inclusion criteria. We found a small but significant reduction in gray matter in the left inferior temporal gyrus for groups of normal hearing individuals with tinnitus compared to groups of hearing-matched individuals without tinnitus. In sharp contrast, in groups with hearing loss, tinnitus was associated with increased gray matter levels in the bilateral lingual gyrus and the bilateral precuneus. Those results were dependent upon matching the hearing levels between the groups with or without tinnitus. The current investigation suggests that hearing loss is the driving force of changes in cortical gray matter across individuals with and without tinnitus. Future studies should carefully account for confounders, including hearing loss, hyperacusis, anxiety, and depression, to identify gray matter changes specifically related to tinnitus. Ultimately, the aggregation of standardized individual datasets with both anatomical and useful phenotypical information will permit a better understanding of tinnitus-related gray matter differences, the effects of potential comorbidities, and their interactions with tinnitus.

## 1. Introduction

Subjective tinnitus (here referred to as simply tinnitus) is the perception of sounds in the ears or head in the absence of an external stimulus. It is a common condition with a prevalence of about 5–40%, depending on the studied population and definition [1,2,3,4,5]. Tinnitus commonly occurs with hearing loss, and the prevalence of tinnitus increases with age [6]. While the majority of the population reports slight to mild tinnitus, millions struggle with severe tinnitus, with a prevalence of about 1–5% [6,7]. Severe tinnitus is associated with a significant impairment in quality of life and occurs in conjunction with anxiety, depression, distress, frustration, hyperacusis, mood swings, poor concentration, and sleep disturbances [1,2,3,4,5,6,7].

Despite its high prevalence and decades of clinical research, the pathophysiology of tinnitus is not well understood. Neuroimaging studies (both functional and anatomical) have revealed changes associated with tinnitus in both auditory and non-auditory pathways. To date, structural neuroimaging studies using voxel-based morphometry (VBM) to investigate tinnitus have reported structural abnormalities in a variety of spatially distinct gray matter regions, including the auditory network (inferior colliculus, medial geniculate body, and primary or secondary auditory cortices), the default mode network (parietal cortices and prefrontal cortices), the salience network (anterior cingulate cortex and anterior insula), the limbic network (accumbens nucleus, amygdala, hippocampus, and parahippocampus), the visual network (primary or secondary visual cortices), the sensorimotor network (supplementary motor area), and the cerebellum [8,9,10,11]. In addition, VBM studies investigating tinnitus have also explored the relationships between these gray matter regions and clinical conditions that occur in conjunction with tinnitus, e.g., anxiety, depression, hearing loss, hyperacusis, and tinnitus burden [12,13,14]. However, the results have been highly inconsistent and sometimes contradictory. The two main reasons for these discrepancies include the heterogeneity of the demographic factors (including the selection and characterization of groups, as well as the relatively small group sizes in general) and methodological factors (including the data analyses and statistical thresholds used) [10,15].

A previous systematic review by Scott-Wittenborn et al. of VBM studies investigating tinnitus was unable to identify robust conclusions across studies [10]. This study reported that there were differences in the selection and characterization of groups with or without tinnitus across studies, making a direct comparison very difficult. In addition, the authors suggested that there is an increasing need to standardize the use of VBM to study gray matter differences associated with tinnitus.

Furthermore, a recent coordinate-based meta-analysis by Cheng et al. of VBM studies investigating tinnitus included eight VBM studies comparing gray matter differences between groups of individuals with tinnitus (*n* = 242) and groups of individuals without tinnitus (*n* = 217) using Anisotropic Effect Size Seed-Based d Mapping (AES-SDM) [15]. This meta-analysis reported significantly increased gray matter in the angular gyrus and the middle–superior temporal gyrus in the group with tinnitus compared to the group without tinnitus. In addition, the gray matter was significantly reduced in the caudate nucleus, the superior frontal gyrus, and the supplementary motor area in the group with tinnitus. This meta-analysis concluded that cortico-striatal circuits are involved in the pathophysiology of tinnitus. However, there were a few important limitations in this study. First, this meta-analysis combined both normal hearing as well as hearing loss groups with or without tinnitus, meaning it did not consider the possible effects of hearing loss. Second, this meta-analysis was thresholded at an uncorrected level of *p* ≤ 0.005, which might be too liberal [16]. Lastly, this meta-analysis used only *x, y, z*-coordinates, although combining unthresholded statistical images with *x, y, z*-coordinates could have improved the sensitivity and specificity of the meta-analysis [16].

To address these limitations, in the current study we conducted a combined image- and coordinate-based meta-analysis of whole-brain VBM studies investigating tinnitus. Using this approach, we aimed to identify robust gray matter differences associated with tinnitus, as well as examining the possible effects of hearing loss on the outcome of the meta-analysis.

## 2. Materials and Methods

### 2.1. Search Strategy

The PubMed and Web of Science databases were searched for studies published up to August 2021 with no restriction on the publication period. The following search terms were used: (1) “tinnitus”[All Fields] AND (“voxel-based morphometry”[All Fields] OR “voxel wise”[All Fields] OR “voxel-based”[All Fields]); and (2) “tinnitus”[All Fields] AND “gray matter”[All Fields] AND (“magnetic resonance imaging”[All Fields] OR “MRI”[All Fields]). To ensure that relevant papers were not missed, a second search was conducted in PubMed and Web of Science databases, replacing the term “gray matter” with “brain volume”: “tinnitus”[All Fields] AND “brain volume”[All Fields] AND (“magnetic resonance imaging”[All Fields] OR “MRI”[All Fields]). This search did not retrieve previously unidentified papers. Additional manual searches were conducted using BrainMap, Neurosynth, and NeuroVault websites for studies published up to December 2021. Finally, previous systematic reviews and meta-analyses were checked to verify that other English-language studies were not missed. The current meta-analysis followed PRISMA guidelines to identify studies [17,18], as illustrated in the PRISMA flowchart in Figure 1.

### 2.2. Study Selection

Studies were considered for inclusion if they (1) were published in a peer-reviewed journal in English; (2) investigated gray matter differences between a group with tinnitus (and either normal hearing or hearing loss) and a group without tinnitus (and either normal hearing or hearing loss); (3) involved a whole-brain VBM analysis; and (4) reported significant results as Montreal Neurological Institute (MNI) or Talairach (TAL) *x, y, z*-coordinates or indicated that significant results were not found. In addition, the corresponding authors were contacted via email by two investigators (P.M. and P.V.D.) to obtain unthresholded statistical images of the whole-brain VBM analysis. If two studies used similar datasets, then only the study with the larger dataset was included to avoid data duplication.

Studies were excluded if they (1) investigated gray matter differences associated with objective tinnitus, e.g., pulsatile tinnitus; (2) did not include a control group without tinnitus (and either normal hearing or hearing loss); (3) did not include a whole-brain VBM analysis or limited their analysis to specific region-of-interests; and (4) did not report significant results as MNI or TAL *x, y, z*-coordinates.

### 2.3. Data Extraction

The following data from each study were extracted and included in an Excel spreadsheet: (1) the study information; (2) the demographic, audiometric, and questionnaire information; (3) the scanning information; (4) the methodological information; (5) the whole-brain VBM results (Appendix A).

### 2.4. Quality Assessment

The quality of the included studies was assessed using a 20-point checklist based on previous neuroimaging meta-analyses [19,20], which were subsequently modified to reflect important variables for the current meta-analysis (Appendix A). To improve the reliability and minimize errors or bias, the study selection, data extraction, and quality assessment were performed independently by two investigators (M.T. and P.M.). Disagreements were resolved by a third investigator (P.v.D.).

The current meta-analysis has been preregistered with the International Prospective Register of Systematic Reviews of the University of York (PROSPERO registration number = CRD42020208248; https://www.crd.york.ac.uk/prospero/, accessed on 10 June 2022).

### 2.5. Combined Image- and Coordinate-Based Meta-Analysis

A combined image- and coordinate-based meta-analysis was performed using Seed-Based d Mapping with Permutation of Subject Images (SDM-PSI version 6.22; https://www.sdmproject.com/, accessed on 10 June 2022). Before preprocessing, input data were prepared for each study as recommended [21]. The detailed preprocessing steps for a combined image- and coordinate-based meta-analysis using SDM-PSI have been described previously [16,21]. Firstly, images of the lower and upper bounds of unbiased possible effect sizes for each study were estimated within a gray matter mask by means of an anisotropic Gaussian kernel (full-width half-maximum (FWHM) = 20 mm; anisotropy = 1; voxel size = 2 × 2 × 2 mm^3^). SDM-PSI uses multiple imputations (imputations = 50) of a Meta-Analysis of Studies with Non-Statistically Significant Unreported Effects (MetaNSUE) algorithms to estimate images of unbiased possible effect sizes for each study, and then adds the standard error [22]. Secondly, images of unbiased possible effect sizes from different imputations were combined using Rubin’s rules to obtain combined images for each study [16]. The combined images were then meta-analyzed using a standard random effects model considering the sample size, inter-study variability, and between-study heterogeneity [16]. Thirdly, the results corrected for family-wise error (FWE) were calculated via the permutation of subject images (number of permutations = 500) [16]. Due to a software limitation, it was not possible to compute the FWE correction with more than 500 permutations (although applying 1000 permutations is more common in the field). Importantly, increasing the number of permutations would only narrow the null distribution, making the current finding even more significant. Thus, the findings reported here would remain unchanged.

Finally, to identify robust gray matter differences associated with tinnitus, as well as to examine the possible effects of hearing loss on the outcome of the meta-analysis, four whole-brain meta-analyses were conducted. The first meta-analysis included all studies retrieved by the online search. The second analysis excluded one study that did not match the groups for hearing level. The third and fourth analyses looked separately at gray matter changes associated with tinnitus in studies that included either individuals with normal audiograms or with hearing loss. This approach resulted in distinguishing four possible groups of individuals: individuals with (1) normal hearing with tinnitus (NH+T), (2) normal hearing without tinnitus (NH−T), (3) hearing loss with tinnitus (HL+T), and (4) hearing loss without tinnitus (HL−T). The results of the whole-brain meta-analyses were corrected for multiple comparisons at the cluster level (FWE *p* ≤ 0.05) using threshold-free cluster enhancement (TFCE) with an initial cluster size of *k* ≥ 10 voxels.

### 2.6. Heterogeneity and Publication Bias

In cases of significant results, the effect sizes (Hedge’s *g*) for each significant cluster were estimated for the included studies using SDM-PSI. The heterogeneity of the results was then calculated using the *I*^2^ statistical test, in which *I*^2^ > 50% is commonly agreed to indicate serious heterogeneity. In addition, Egger’s test was used to assess the asymmetry of the funnel plot by examining the publication bias, where Egger’s *p* ≤ 0.05 indicates a significant publication bias. These analyses were performed using R version 4.1.3 (R Core Team, 2022; https://www.R-project.org/, accessed on 10 June 2022) with the metafor package (https://www.metafor-project.org/, accessed on 10 June 2022).

## 3. Results

### 3.1. Studies Included

Of the 153 identified studies (see Figure 1), a total of 15 studies met the inclusion criteria, resulting in the inclusion of a total of 423 individuals with tinnitus and either normal hearing or hearing loss (mean age 50.94 years; 173 females) and 508 individuals without tinnitus and either normal hearing or hearing loss (mean age 51.59 years; 234 females) [13,23,24,25,26,27,28,29,30,31,32,33,34,35,36]. Unthresholded statistical images of whole-brain VBM analyses were obtained for 5 studies [23,27,31,32,36]. The demographic, audiometric, questionnaire, scanning, and methodological information as well as the whole-brain VBM results are shown in Table 1. More detailed information is provided in Appendix A.

### 3.2. SDM-PSI Meta-Analyses

#### 3.2.1. Effect of Tinnitus (All Studies Included)

A total of 15 studies [13,23,24,25,26,27,28,29,30,31,32,33,34,35,36] compared groups of individuals with tinnitus (NH+T and HL+T: *n* = 423; mean age 50.94 years; 173 females) to groups of individuals without tinnitus (NH−T and HL−T: *n* = 508; mean age 51.59 years; 234 females) (Table 1). These 15 studies either matched or did not match their groups for hearing loss (Table 1). The whole-brain meta-analysis showed that there was no significantly increased gray matter in the tinnitus group compared to the controls. However, tinnitus in this analysis was associated with significantly reduced gray matter in the left anterior cingulate cortex (with a cluster extending to the medial prefrontal cortex), the left interior temporal gyrus, and the left middle temporal gyrus (Table 2A; Figure 2A). 

#### 3.2.2. Effect of Tinnitus When Groups Are Matched for Hearing Levels (One Study Excluded)

One of the 15 retrieved studies did not match groups in terms of hearing levels [31]. Furthermore, this study intentionally included individuals with significant anxiety or depression (in both the tinnitus and the control groups). Therefore, in a second meta-analysis, in which this study was excluded (NH+T and HL+T vs. NH−T and HL−T excluding study [31]), we investigated the effect of tinnitus when the groups were matched for hearing levels. The 14 studies [13,23,24,25,26,27,28,29,30,32,33,34,35,36] included in this analysis compared 364 individuals with tinnitus (NH+T and HL+T excluding study [31]: mean age 50.96 years; 146 females) and 320 individuals without tinnitus (NH−T and HL−T excluding study [31]: mean age 51.86 years; 146 females) (Table 1). The results of the whole-brain meta-analysis showed that when matched for hearing levels, the groups with tinnitus had significantly higher gray matter levels in the right cuneus, the bilateral lingual gyrus (with a cluster extending to the cerebellum), the right posterior cingulate cortex, and the left precuneus compared to the groups without tinnitus (Table 2B; Figure 2B). No regions with significantly lower gray matter in the groups with tinnitus were found in this analysis

#### 3.2.3. Effect of Tinnitus in Groups with Normal Hearing

Seven of the 15 studies [23,24,28,29,33,35,36] compared groups of normal hearing individuals with tinnitus (NH+T: *n* = 168; mean age 43.36 years; 81 females) and groups of normal hearing individuals without tinnitus (NH−T: *n* = 166; mean age 43.63 years; 84 females) (Table 1). The meta-analysis of these studies revealed a small area of decreased gray matter in the left inferior temporal gyrus in the group with tinnitus compared to the group without tinnitus (Table 3A; Figure 3A).

#### 3.2.4. Effect of Tinnitus in Groups with Hearing Loss

Seven of the 15 studies [13,25,26,27,30,32,34] compared groups of hearing loss individuals with or without tinnitus (HL+T: *n* = 138; mean age 57.02 years; 44 females and HL−T: *n* = 114; mean age 59.14 years; 46 females) (Table 1). In all seven studies, the groups were well matched in terms of hearing levels. A whole-brain meta-analysis of these studies showed that the group with hearing loss and tinnitus had significantly increased gray matter levels in the bilateral lingual gyrus (with a cluster extending in the precuneus and the cerebellum) and the left precuneus (with a cluster extending in the lingual gyrus) (Table 3B; Figure 3B) compared to the group with hearing loss but without tinnitus.

#### 3.2.5. Heterogeneity and Publication Bias

The approximative effect sizes (Hedge’s *g*), test of heterogeneity (*I*^2^) results, and Egger’s probabilities of publication bias are provided for every significant group difference in Table 2 and Table 3. The effect sizes were modest. The heterogeneity across studies was surprisingly low. Figure 4 shows the funnel plots for every significant finding. Egger’s tests of funnel plot asymmetry were not significant for any of the findings across the four meta-analyses (all *p* > 0.05), suggesting no publication bias.

## 4. Discussion

The current study is the first combined image- and coordinate-based meta-analysis of VBM studies to examine gray matter differences associated with tinnitus and hearing loss. We found that matching for hearing levels affected the results of the meta-analyses. Depending on matching for hearing levels, we identified several areas with gray matter differences associated with tinnitus (including decreased gray matter in the left anterior cingulate and medial prefrontal cortex, decreased gray matter in the left inferior and middle temporal gyrus, and increased gray matter in the bilateral lingual gyrus and in the bilateral precuneus). These findings provide new insights into brain changes driven by tinnitus and hearing loss. Importantly, our findings support previous observations indicating that the major driver of gray matter changes in the brain is not tinnitus but rather hearing loss and aging [32]. Our findings, moreover, motivate recommendations for future VBM studies of tinnitus.

### 4.1. Effects of Matching Hearing Levels When Identifying Changes in Gray Matter Associated with Tinnitus

In this study, we conducted a series of four meta-analyses. The first analysis included all (15) retrieved studies and revealed three clusters of decreased gray matter in individuals with tinnitus compared to individuals without tinnitus (regardless of hearing level, i.e., NH+T and HL+T vs. NH−T and HL−T). One large cluster covered parts of the anterior cingulate cortex and medial prefrontal cortex. Two small clusters of decreased gray matter were further observed in the left inferior and middle temporal gyri (Table 2A; Figure 2A). No regions with significantly higher gray matter in individuals with tinnitus were found in this analysis.

To more carefully control for the hearing level in the meta-analysis, we performed a second meta-analysis excluding the one study [31] that did not match groups of individuals for hearing levels and included individuals with clinically significant depression and anxiety. In stark contrast to the result of the first meta-analysis, this second meta-analysis (NH+T and HL+T vs. NH−T and HL−T excluding study [31]), in which the hearing levels were now matched, found no areas of decreased gray matter in the individuals with tinnitus compared to the individuals without tinnitus. Moreover, several clusters of increased grey matter were found in the individuals with tinnitus. The largest clusters of increased gray matter were observed in the bilateral lingual gyrus with extensions into the cerebellum. Other smaller clusters were found in the right cuneus, the right posterior cingulate, and left precuneus (Table 2B; Figure 2B). 

In light of these results, the third and fourth meta-analyses were conducted to test whether these clusters of gray matter changes in individuals with tinnitus compared to individuals without tinnitus were observed when specifically comparing subgroups with and without hearing loss. The third analysis (comparing NH+T vs. NH−T) found that in studies that included only individuals with normal hearing (*n* = 7 studies), the only difference between the individuals with and without tinnitus was a small area of decreased gray matter in the left inferior temporal gyrus of those with tinnitus (Table 3A; Figure 3A). The fourth and final analysis (comparing HL+T vs. HL−T), investigating the effects of tinnitus in groups of individuals with hearing loss (*n* = 7 studies), uncovered a series of clusters showing increased gray matter in individuals with tinnitus compared to individuals without tinnitus. These clusters of increased gray matter were again found in the bilateral lingual gyrus extending into the precuneus and the cerebellum (Table 3B; Figure 3B).

The combined interpretation of these four meta-analyses indicates that matching for hearing levels dramatically affects the results. In particular, in our results, hearing loss was the driving force of changes in cortical gray matter across individuals with and without tinnitus. These findings disentangle the brain regions important in the pathology of tinnitus and hearing loss (see Section 4.2) and also motivate recommendations for future VBM studies of tinnitus (see Section 4.3).

The results of our investigation are not consistent with a recent meta-analysis that included eight VBM studies and also used Seed-Based d Mapping (AES-SDM) like our study [15]. The authors of this study reported significantly reduced gray matter levels in the caudate nucleus, the medial superior frontal gyrus, and the supplementary motor area, as well as increased gray matter in the angular and the middle temporal gyri in the groups with tinnitus compared to the groups without tinnitus [15]. Several reasons could explain the variability in findings between our study and this previous study. First, this previous study [15] did not distinguish between studies including individuals with and without hearing loss, which we found greatly impacted the results. Secondly, there were differences in study selection criteria (specifically, we did not include one study in the Chinese language). Finally, the validity of a meta-analysis is heavily dependent on the precision of the data under review. With the currently available studies, meta-analyses cannot control for the effects of important confounders like hyperacusis, anxiety, and depression, all know to be commonly associated with tinnitus and tinnitus burden [8,9,10]. Furthermore, the sample sizes are on average quite small and the employed image preprocessing methods, statistical models, and significance thresholds vary wildly across studies [10].

### 4.2. Brain Regions Associated with Tinnitus and Hearing Loss

In the four meta-analyses performed in this study, we identified various brain regions with significant changes in gray matter associated with tinnitus and hearing loss. Specifically, we found a significant (1) decrease in gray matter in the anterior cingulate and medial prefrontal cortex in individuals with tinnitus (when not specifically matching for hearing loss, NH+T and HL+T vs. NH−T and HL−T); (2) decrease in gray matter in the inferior and middle temporal gyri in individuals with tinnitus (when comparing groups with normal hearing, NH+T vs. NH−T); and (3) increase in gray matter in the lingual gyrus and the precuneus in individuals with tinnitus (when comparing groups with hearing loss, HL+T vs. HL−T).

Our first key significant finding indicates decreased gray matter in the anterior cingulate and medial prefrontal cortex in participants with tinnitus (when not specifically matching for hearing loss, NH+T and HL+T vs. NH−T and HL−T). The anterior cingulate cortex is connected to the “cognitive” prefrontal cortex, the “emotional” limbic network, and the "sensory” auditory and visual networks [37,38,39]. Therefore, the anterior cingulate cortex plays a crucial role in integrating the neural circuits for affect regulation and sensory attention [37,38,39]. Affect regulation refers to the mechanisms by which unpleasant emotions are modulated or suppressed to achieve homeostasis, and is an important concept in psychology regarding emotional regulation [37,38,39]. Similarly, the anterior cingulate cortex has been described as a key region of the salience network in the context of tinnitus [40,41]. The salience network is involved in modulating or suppressing unpleasant auditory stimuli and identifying relevant auditory stimuli for attention [40,41,42]. Thus, significantly reduced gray matter in the anterior cingulate cortex (with a cluster extending in the medial prefrontal cortex) might be related to a lack of modulation or suppression of unpleasant auditory stimuli in individuals with tinnitus and either normal hearing or hearing loss. This result is in agreement with a previous meta-analysis that found reduced gray matter in the frontal lobe areas in individuals with tinnitus and either normal hearing or hearing loss compared to individuals without tinnitus and either normal hearing or hearing loss [15].

Both the anterior cingulate cortex and the medial prefrontal cortex have been previously found to be important in the emotional aspects of tinnitus [42]. A previous magnetoencephalography study reported that the functional connectivity between the anterior cingulate cortex and prefrontal cortex correlated negatively with the tinnitus burden [43]. Furthermore, the inhibitory low-frequency repetitive transcranial magnetic stimulation of the anterior cingulate cortex–medial prefrontal cortex using a double cone coil has been investigated as an innovative approach for the treatment of tinnitus, and has been shown to reduce the tinnitus burden [44]. These findings suggest a link between the tinnitus burden and hyperactivity of the anterior cingulate–medial prefrontal cortex. Future research is needed to clarify the role of these brain regions in the assessment and treatment of tinnitus, and especially regarding the tinnitus burden.

Our second key significant finding indicates decreased gray matter levels in a small area of the left inferior temporal gyrus in individuals with tinnitus (when comparing groups with normal hearing, NH+T vs. NH−T). The same finding was also observed when not specifically matching for hearing loss (i.e., NH+T and HL+T vs. NH−T and HL−T). The temporal lobe plays an important role in various cognitive functions, including auditory processing and language or speech comprehension [45]. Several studies have reported impairments in these domains in individuals with tinnitus [46,47,48,49,50]. Thus, our finding of reduced gray matter in the inferior temporal gyrus in tinnitus is not surprising. One study investigated speech perception in normal hearing individuals with tinnitus using a Mandarin version of the speech perception in noise test [51]. Compared to the normal hearing group without tinnitus, the normal hearing group with tinnitus displayed lower scores in perceiving sentences in a noisy environment [51]. Future studies in normal hearing individuals with tinnitus should explore the possible correlation between temporal lobe gray matter levels and scores on the speech in noise test. 

Our third key significant finding indicates increased gray matter in individuals with tinnitus (when comparing groups with hearing loss, HL+T vs. HL−T). This difference was most prominent in the bilateral lingual gyrus and the bilateral precuneus. The same regions were found in the meta-analysis that included all studies that had matched their groups for hearing levels (i.e., NH+T and HL+T vs. NH−T and HL−T excluding study [31]). These results are in agreement with the previous conclusion that whereas hearing loss is associated with the loss of gray matter, hearing loss with the addition of tinnitus preserves gray matter [25,32]. In particular, the lingual gyrus and the precuneus are better preserved in individuals with hearing loss and tinnitus.

The lingual gyrus plays a role in encoding visual memories and is directly connected to the limbic network [52]. Damage to this region may lead to visual memory dysfunction [53]. In addition, previous studies have reported that the lingual gyrus is associated with better performance on memory tasks and cognitive functions [54,55,56]. Thus, our results suggest that hearing loss with additional tinnitus may be linked to better cognitive function than hearing loss without tinnitus. Supporting this conclusion, a recent cross-sectional study aimed at delineating the interaction between tinnitus and cognition using a composite z-score from four cognitive tests found that in (non-Hispanic) groups with hearing loss, the additional presence of tinnitus was associated with better cognitive performance compared to the group without tinnitus [57]. Future studies are needed to investigate the contribution of the lingual gyrus (and other bran regions) to preserved cognitive performance in individuals with hearing loss with and without tinnitus.

The precuneus is part of the default mode network and plays an essential role in various complex cognitive functions, including conscious and internal awareness [58]. The default mode network is most active at wakeful rest and is functionally anticorrelated with the dorsal attention network [59]. A previous resting-state functional imaging study reported that tinnitus is associated with decreased functional connectivity within the default mode network and increased functional connectivity between the default mode network and the dorsal attention network [60]. Similarly, a previous diffusion imaging study, reporting reduced connectivity in the precuneus, speculated that the constant disturbance of tinnitus during wakeful rest may stimulate structural changes in the precuneus [61]. Future work is needed to investigate the mechanistic links between changes in attentional networks associated with hearing loss and additional tinnitus and preserved gray matter in the precuneus of individuals with hearing loss and additional tinnitus.

### 4.3. Recommendations for Future VBM Studies of Tinnitus

Our study identifies the importance of matching individuals for hearing levels when investigating changes in gray matter associated with tinnitus. Our study also suggests that age and psychiatric comorbidities may influence the results. To investigate the interactions between tinnitus, hearing loss, age, anxiety, and depression when investigating brain changes associated with tinnitus, future VBM studies should carefully document the various demographic and methodological factors outlined in Table 4. Documenting comorbidities, such as depression and anxiety, is especially important, since it is not always feasible or desirable to exclude these groups. In addition, future studies should investigate the effects of the duration of tinnitus and the cause or duration of hearing loss on changes in brain volume. Two previous studies have researched the impacts of the tinnitus duration on gray matter levels in a larger tinnitus population [12,62]. To date, no tinnitus study has examined gray matter changes in response to the cause or duration of hearing loss (although the onset of hearing loss is often difficult to identify). Considering that our current results show that hearing loss is the main factor driving changes in gray matter, future research investigating the aspects of hearing loss that drive these changes is important. Finally, the effect of sex should also be considered in VBM studies. Here, we specifically recommend matching for the sex ratio when designing studies. We also recommend including the total intracranial volume (TIV), which is on average 12% larger in males [63] and can account for variations within as well as between sexes. We do not, however, recommend including both sex and TIV as covariates, since they are significantly correlated. We believe the best practical solution currently available is to match sex ratios between groups and include TIV as a covariate to control for interindividual variability in brain size or gray matter across sexes. However, if it is suspected that sex would influence the effect of the tinnitus or hearing loss on the regional gray matter, it would be important to design specific studies including sex, not as a confounder, but as a grouping variable.

VBM studies should, furthermore, upload the unthresholded statistical images to a shared repository (e.g., NeuroVault, https://neurovault.org/, accessed on 10 June 2022). A concerted effort across researchers to curate a repository with a large number of individual data sets that include robust phenotypic data and T1 images is necessary to distinguish the brain changes associated specifically with tinnitus from the effects of hearing loss, age, hyperacusis, anxiety, and depression, and to investigate the interactions between these variables. We suspect that these recommendations are equally valid and important for other brain imaging approaches.

## Figures and Tables

**Figure 1 brainsci-12-01192-f001:**
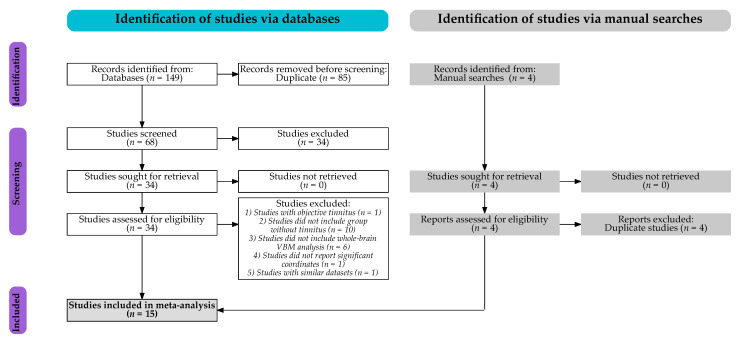
The PRISMA flowchart for studies included in the current meta-analysis.

**Figure 2 brainsci-12-01192-f002:**
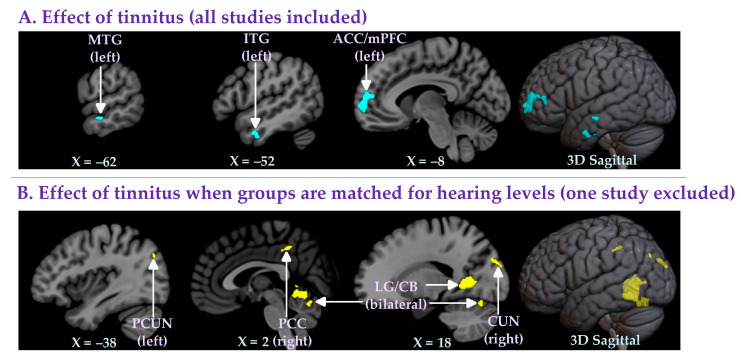
Whole-brain meta-analyses of studies comparing individuals with and without tinnitus, (**A**) including all studies and (**B**) excluding one study [31] that did not match groups for hearing levels, using TFCE FWE-corrected statistics (*p* ≤ 0.05, cluster size *k* ≥ 10 voxels). (**A**) Lower gray matter levels in the left anterior cingulate cortex (with a cluster extending to the medial prefrontal cortex), the left inferior temporal gyrus, and the left middle temporal gyrus in the groups with tinnitus (NH+T and HL+T) compared to the groups without tinnitus (NH−T and HL−T), regardless of whether or not the groups were matched for hearing levels. (**B**) Higher gray matter levels in the right cuneus, the bilateral lingual gyrus (with a cluster extending to the cerebellum), the right posterior cingulate cortex, and the left precuneus in the groups with tinnitus (NH+T and HL+T excluding study [31]) compared to the groups without tinnitus (NH−T and HL−T excluding study [31]) and matched for hearing levels. Note: ACC, anterior cingulate cortex; CB, cerebellum; CUN, cuneus; ITG, inferior temporal gyrus; LG, lingual gyrus; mPFC, medial prefrontal cortex, MTG, middle temporal gyrus; PCC, posterior cingulate cortex; PCUN, precuneus.

**Figure 3 brainsci-12-01192-f003:**
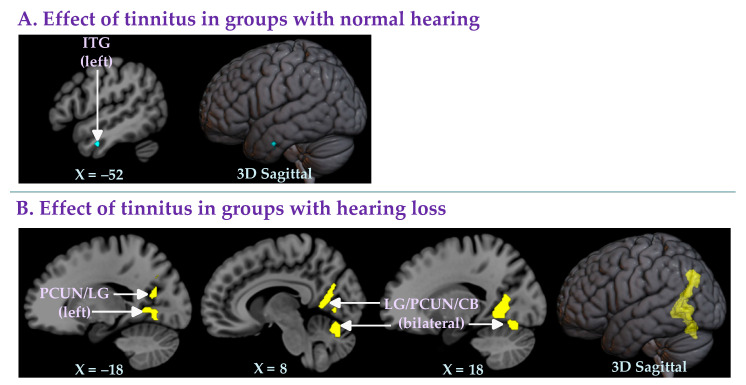
Whole-brain meta-analyses of studies comparing individuals with tinnitus and (**A**) normal hearing or (**B**) hearing loss, using TFCE FWE-corrected statistics (*p* ≤ 0.05, cluster size *k* ≥ 10 voxels). (**A**) Lower gray matter levels in the left inferior temporal gyrus in the groups with normal hearing and tinnitus (NH+T) compared to the groups with normal hearing but without tinnitus (NH−T). (**B**) Higher gray matter levels in the bilateral lingual gyrus (with a cluster extending to the precuneus and the cerebellum) and the left precuneus (with a cluster extending to the lingual gyrus) in the groups with hearing loss and tinnitus (HL+T) compared to the groups with hearing loss but without tinnitus (HL−T). Note: CB, cerebellum; ITG, inferior temporal gyrus; LG, lingual gyrus; PCUN, precuneus.

**Figure 4 brainsci-12-01192-f004:**
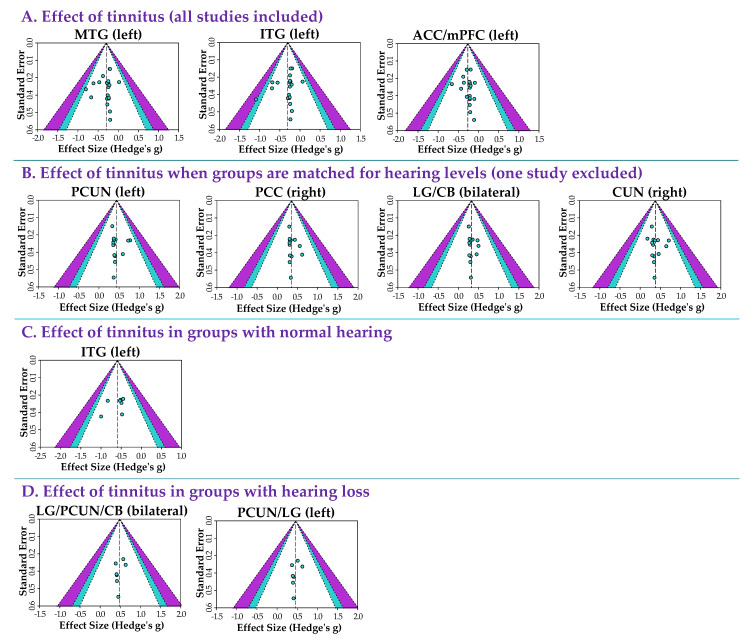
Heterogeneity and publication bias. Funnel plots for every significant finding obtained across the four meta-analyses. None of the funnel plots showed evidence of a reporting publication bias (Egger’s tests: all *p* > 0.05).

**Table 1 brainsci-12-01192-t001:** Demographic, audiometric, questionnaire, scanning, and methodological information as well as whole-brain VBM results for studies included in the current meta-analysis (see Appendix A for detailed information).

Study	Groups	*n* (Females)	Mean Age (years)	Groups Matched for			Mean Tinnitus Duration (months)	Mean Tinnitus Severity Score	MRI Scanner	Smoothing FWHM	Significant Threshold	Whole-Brain VBM Results Associated with Tinnitus
				Age	Sex	HL						
Mühlau et al. [23]	NH+T	28 (15)	40	Yes	Yes	Yes	53 ± 52	TQ = 25	1.5T	8	FDR *p* < 0.05 voxel-level	↑GM: NS
	NH−T	28 (15)	39				-	-				↓GM: SCG
Landgrebe et al. [24]	NH+T	28 (13)	32.2	Yes	Yes	Yes	53.3	TQ = 32.9	1.5T	10	FDR *p* < 0.05 voxel-level	↑GM: NS
	NH−T	28 (13)	31.2				-	-				↓GM: NS
Husain et al. [25]	HL+T	8 (0)	56.13	Yes	Yes	No	NA	THI = 17.25	3T	8	Uncorrected *p* < 0.001	↑GM: NS
	NH−T	11 (0)	48.09				-	-				↓GM: NS
	HL+T	8 (0)	56.13	Yes	Yes	Yes	NA	THI = 17.25				↑ GM: MFG, SFG
	HL−T	7 (0)	51.38									↓GM: NS
Leaver et al. [26]	HL+T	23 (11)	47.4	Yes	Yes	Yes	149.9	NA	3T	6	Uncorrected *p* < 0.002	↑GM: NS
	HL−T	21 (13)	49				-	-				↓GM: dmPFC, SMG, vmPFC
Boyen et al. [27]	HL+T	31 (11)	56	No	NA	No	NA	THI = 29	3T	8	FWE *p* < 0.05 voxel-level	↑GM: MTG, SMG, STG
	NH−T	24 (8)	58				-	-				↓GM: HTH, OG, SFG
	HL+T	31 (11)	56	No	NA	Yes	NA	THI = 29				↑GM: NS
	HL−T	16 (3)	63				-	-				↓GM: NS
Melcher et al. [28]	NH+T	24 (12)	46.9	Yes	Yes	Yes	NA	TRQ = 26.7	3T	8	Uncorrected *p* < 0.001	↑GM: NS
	NH−T	24 (12)	45.8				-	-				↓GM: NS
Allan et al. [29]	NH+T/HL+T	73 (30)	58.38	Yes	Yes	Yes	NA	NA	1.5T, 3T	10	FWE *p* < 0.05	↑GM: NS
	NH−T/HL−T	55 (25)	56.91				-	-				↓GM: NS
	NH+T	15 (9)	47.6	Yes	Yes	Yes	NA	NA				↑GM: NS
	NH−T	15 (9)	50.2				-	-				↓GM: NS
Schmidt et al. [13]	HL+T	15 (5)	55.13	NA	NA	No	NA	THI = 9.33	3T	10	Uncorrected *p* < 0.001 cluster-level	↑GM: ACC, SFG
	NH−T	13 (6)	52.93				-	-				↓GM: NS
	HL+T	15 (5)	55.13	NA	NA	Yes	NA	THI = 9.33				↑GM: NS
	HL−T	13 (8)	57.61				-	-				↓GM: NS
Luan et al. [30]	HL+T	10 (5)	52	Yes	Yes	No	NA	NA	3T	8	FDR *p* < 0.05 voxel-level	↑GM: NS
	NH−T	35 (18)	55.97				-	-				↓GM: NS
	HL+T	10 (5)	52	Yes	Yes	Yes	NA	NA				↑GM: NS
	HL−T	25 (9)	55.48				-	-				↓GM: NS
Besteher et al. [31]	HL+T	59 (27)	50.6	NA	NA	No	NA	TQ = 42.2	3T	8	Uncorrected *p* < 0.001	↑GM: NS
	NH−T	66 (35)	45				-	-				↓GM: IPC, PHPC, PCUN
Koops et al. [32]	HL+T	39 (6)	59.2	No	NA	No	147.5	THI = 35	3T	8	TFCE FWE *p* < 0.05	↑GM: NS
	NH−T	39 (21)	45.7				-	-				↓GM: NS
	HL+T	39 (6)	59.2	Yes	NA	Yes	147.5	THI = 35				↑GM: LG
	HL−T	22 (9)	62.6				-	-				↓GM: NS
Wei et al. [33]	NH+T	27 (15)	46.4	Yes	Yes	Yes	23.4 -	THI = 40 -	3T	6	FDR *p* < 0.05 cluster-level	↑GM: NS
	NH−T	27 (15)	46.6									↓GM: CN, TH
Lee et al. [34]	HL+T	12 (6)	73.27	Yes	Yes	Yes	42	THI = 38.5	3T	8	Uncorrected *p* < 0.005	↑GM: NS
	HL−T	11 (4)	74.83				-	-				↓GM: INS
Wei et al. [35]	NH+T	33 (10)	48.2	Yes	Yes	Yes	NA	THI = 52.5	3T	6	FDR *p* < 0.05 cluster-level	↑GM: NS
	NH−T	26 (11)	47.3				-	-				↓GM: CAU, CAL, CUN, PHPC, STG
Chen et al. [36]	NH+T	13 (7)	42.23	Yes	Yes	Yes	NA	THI = 53.38	3T	8	FWE *p* < 0.001 cluster-level	↑GM: NS
	NH−T	18 (9)	45.33				-	-				↓GM: MTG

Note: HL+T, hearing loss group with tinnitus; HL−T, hearing loss group without tinnitus; NH+T, normal hearing group with tinnitus; NH−T, normal hearing group without tinnitus; THI, tinnitus handicap inventory; TQ, tinnitus questionnaire; TRQ, tinnitus reaction questionnaire; ↑GM, increased gray matter; ↓GM, decreased gray matter; ACC, anterior cingulate cortex; CAL, calcarine gyrus; CAU, caudate nucleus; CN, cochlear nucleus; CUN, cuneus; dmPFC, dorsomedial prefrontal cortex; HTH, hypothalamus; INS, insula; IPC, inferior parietal cortex; LG, lingual gyrus; OG, occipital gyrus; MFG, medial frontal gyrus, MTG, middle temporal gyrus; PCUN, precuneus; PHPC, parahippocampus; SCG, subcallosal gyrus; SFG, superior frontal gyrus; SMG, supramarginal gyrus; STG, superior temporal gyrus; TH, thalamus; vmPFC, ventromedial prefrontal cortex; NA, not available; NS, not significant.

**Table 2 brainsci-12-01192-t002:** Whole-brain meta-analyses of studies comparing individuals with and without tinnitus.

Gray Matter Regions	MNI *X, Y, Z*	SDM *Z*	FWE *p* TFCE	Voxels	Hedge’s *g* Cluster	*I*^2^ Cluster	Egger’s *p* Cluster
**A. Effect of tinnitus (all studies included) ^a^**							
NH+T and HL+T > NH−T and HL−T							
NS	-	-	-	-	-	-	-
NH+T and HL+T < NH−T and HL−T							
Left anterior cingulate cortex/medial prefrontal cortex (ACC/mPFC)	−12, 44, 8	−4.10	0.028	224	−0.27	0 %	0.787
	−8, 56, 10	−4.01	0.018				
	−16, 66, −6	−3.63	0.036				
	−8, 62, 0	−3.60	0.026				
Left inferior temporal gyrus (ITG)	−54, −10, −34	−3.80	0.036	45	−0.30	0 %	0.392
	−54, −6, −34	−3.69	0.036				
	−56, −16, −32	−3.56	0.046				
Left middle temporal gyrus (MTG)	−62, −16, −14	−4.06	0.036	20	−0.30	0 %	0.614
	−60, −12, −12	−3.85	0.038				
**B. Effect of tinnitus when groups are matched for hearing levels (one study excluded) ^b^**							
NH+T and HL+T > NH−T and HL−T excluding study [31]							
Right cuneus (CUN)	16, −94, 24	4.16	0.016	67	0.38	0 %	0.738
	18, −88, 30	3.90	0.028				
Bilateral lingual gyrus/cerebellum (LG/CB)	20, −52, 2	4.39	0.002	948	0.33	0 %	0.789
	−4, −56, −8	4.26	0.006				
	12, −58, 0	3.96	0.004				
	16, −72, −18	3.19	0.016				
	4, −50, 0	3.15	0.012				
	2, −66, −20	3.13	0.020				
	−2, −64, −20	3.09	0.020				
	14, −72, −14	2.99	0.018				
	8, −68, −16	2.98	0.020				
	14, −72, 4	2.92	0.022				
Right posterior cingulate gyrus (PCC)	0, −42, 46	4.24	0.036	28	0.35	0 %	0.724
	0, −32, 42	3.80	0.038				
Left precuneus (PCUN)	−38, −78, 14	4.70	0.038	12	0.44	0 %	0.813
NH+T and HL+T < NH−T and HL−T excluding study [31]							
NS	-	-	-	-	-	-	-

Note: ^a^ including all studies or ^b^ excluding one study that did not match groups for hearing levels. HL+T, hearing loss group with tinnitus; HL−T, hearing loss group without tinnitus; NH+T, normal hearing group with tinnitus; NH−T, normal hearing group without tinnitus; NS, not significant.

**Table 3 brainsci-12-01192-t003:** Whole-brain meta-analyses of studies comparing individuals with and without tinnitus.

Gray Matter Regions	MNI *X, Y, Z*	SDM *Z*	FWE *p* TFCE	Voxels	Hedge’s *g* Cluster	*I*^2^ Cluster	Egger’s *p* Cluster
**A. Effect of tinnitus in groups with normal hearing ^a^**							
NH+T > NH−T							
NS	-	-	-	-	-	-	-
NH+T < NH−T							
Left inferior temporal gyrus (ITG)	−52, −6, −30	−4.12	0.042	12	−0.59	0 %	0.517
**B. Effect of tinnitus in groups with hearing loss ^b^**							
HL+T > HL−T							
Right lingual gyrus/precuneus/cerebellum (LG/PCUN/CB)	18, −56, 0	4.57	< 0.001	931	0.49	0 %	0.735
	18, −60, 10	3.56	0.008				
	4, −68, −20	3.44	0.012				
	8, −70, −18	3.38	0.012				
	18, −70, −16	3.26	0.012				
	8, −66, −24	3.00	0.016				
	28, −66, −10	2.90	0.042				
Left precuneus/lingual gyrus (PCUN/LG)	−14, −68, 38	3.66	0.012	466	0.46	0 %	0.768
	−14, −68, −2	3.21	0.018				
	−16, −66, −8	2.98	0.020				
	−18, −58, 0	2.94	0.020				
	−18, −66, 18	2.85	0.020				
	−12, −58, 8	2.61	0.030				
HL+T < HL−T							
NS	-	-	-	-	-	-	-

Note: ^a^ normal hearing or ^b^ hearing loss. HL+T, hearing loss group with tinnitus; HL−T, hearing loss group without tinnitus; NH+T, normal hearing group with tinnitus; NH−T, normal hearing group without tinnitus; NS, not significant.

**Table 4 brainsci-12-01192-t004:** Recommendations for future (meta-analyses of) VBM studies investigating tinnitus.

**Demographic Factors**
Match groups for age, sex ratio, and hearing loss
Provide clear definitions for hearing loss (cause and duration) or provide audiograms for both ears
Report clinical information (anxiety, depression, hyperacusis, and tinnitus burden scores and tinnitus duration)
**Methodological Factors**
Provide clear description of the software, realignment algorithm, normalized template, resampling resolution, modulation, and smoothing kernel used during preprocessing
Correct results for multiple comparisons at the cluster or voxel level (FWE, FWE-TFCE, or FDR correction)
Control results for age and total intracranial volume
Report significant *x*, *y*, *z*-coordinates with t-values, z-values, and p-values
**Data Sharing**
Make unthresholded statistical images available, e.g., via NeuroVault website (https://neurovault.org/, accessed on 10 June 2022), or create a shared initiative to aggregate a large number of data sets with minimal requirements on T1 brain imaging protocols and questionnaires for controlling for potential variables of interest such as tinnitus duration, depression, and anxiety.

## Data Availability

The datasets analyzed in the current meta-analysis are available from the corresponding author upon reasonable request.

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
