# Peer review of "A Combined Image- and Coordinate-Based Meta-Analysis of Whole-Brain Voxel-Based Morphometry Studies Investigating Subjective Tinnitus"

_brainsci, 2022, doi:10.3390/brainsci12091192_

Round 1
Reviewer 1 Report
It is a very nice paper, methodologically correct.
The argument is very interesting and conclusion are important for future work or clinical applications, i.e.hearing aids utility in prevention or therapy of tinnitus with hearing loss.
The Authors have hypothesized future evaluation about the degree of hearing loss and gray matter modifications but they should also consider the importance of duration of tinnitus and the cause of hearing loss on the appearance of these modifications therefore I suggest to include some words about these aspects in the discussion.
Reviewer 2 Report
The study proposal is very interesting for the scientific and clinical community, the authors presented the results in detail, but the search methodology seemed a little weak regarding the scientific rigor of the search strategies used in meta-analysis and systematic review, resulting in few studies that decrease the power to infer conclusive results or consistent.
I did not find the boolean strategy used to search the studies in each database, the brain morphometric can be reported by many other terms did not use in the search strategy. I suggest the inclusion of other terms for brain volume morphometric analysis, to enhance the paper identification, and the description in detail of the search method. How the search terms were applied in the database through the general exploring, in articles title, in keywords, .....?
In another point of the method section, I would like to know, why the choice of 500 permutations in the FWE correction.
Round 2
Reviewer 2 Report
I am satisfied with the authors' corrections and responses
Author Response
Thank you for your comments and remarks.